# Gene Regulatory Network of the Noncoding RNA *Qrr5* Involved in the Cytotoxicity of *Vibrio parahaemolyticus* during Infection

**DOI:** 10.3390/microorganisms10102084

**Published:** 2022-10-21

**Authors:** Fei Zha, Rui Pang, Shixuan Huang, Jumei Zhang, Juan Wang, Moutong Chen, Liang Xue, Qinghua Ye, Shi Wu, Meiyan Yang, Qihui Gu, Yu Ding, Hao Zhang, Qingping Wu

**Affiliations:** 1School of Food Science and Technology, Jiangnan University, Wuxi 214122, China; 2Guangdong Provincial Key Laboratory of Microbial Safety and Health, Ministry of Agriculture and Rural Affairs, Key Laboratory of Agricultural Microbiomics and Precision Application, State Key Laboratory of Applied Microbiology Southern China, Institute of Microbiology, Guangdong Academy of Sciences, Guangzhou 510070, China

**Keywords:** *Qrr5*, cytotoxicity, transcriptome, weighted co-expression network analysis, *Vibrio parahaemolyticus*, virulence factor

## Abstract

Small non-coding RNAs (sRNAs) in bacteria are important regulatory molecules for controlling virulence. In *Vibrio* spp., Qrr sRNAs are critical for quorum-sensing pathways and regulating the release of some virulence factors. However, the detailed role of Qrr sRNAs in the virulence of *Vibrio* *parahaemolyticus* remains poorly understood. In this study, we identified a *Vibrio* sRNA *Qrr5* that positively regulates cytotoxicity and adherence in Caco-2 cells by primarily regulating the T3SS1 gene cluster. A number of 185, 586, 355, and 74 differentially expressed genes (DEGs) detected at 0, 2, 4, and 6 h post-infection, respectively, were mainly associated with ABC transporters and two-component system pathways. The DEGs exhibited a dynamic change in expression at various time points post-infection owing to the deletion of *Qrr5*. Accordingly, 17 related genes were identified in the co-expression network, and their interaction with *Qrr5* was determined based on weighted co-expression network analysis during infection. Taken together, our results provide a comprehensive transcriptome profile of *V. parahaemolyticus* during infection in Caco-2 cells.

## 1. Introduction

Small non-coding RNAs (sRNAs) serve as indispensable regulators of many bacterial signaling pathway cascades and pathogenesis [1,2,3]. A novel sRNA, *EsrF*, can sense high ammonium concentrations in the colon and facilitate bacterial motility and adhesion to the host cell, thereby promoting the pathogenicity of *Escherichia coli* O157 [4]. Qrr sRNAs are Hfq-dependent trans-encoded sRNAs involved in the regulation of quorum sensing in *Vibrio* spp. [5,6,7], which positively regulate the expression of *AphA*, a low-cell-density core regulator that suppresses the expression of *LuxR*, a high-cell-density core regulator [8,9,10]. In *Vibrio harveyi*, the *Qrr3* sRNA suppresses *LuxR* through catalytic degradation, *LuxM* through degradation, and *LuxO* through a blockade, as well as activating *AphA* by exposing the ribosome-combining site, whereas the sRNA itself is decomposed [11,12]. However, five RNAs (*Qrr1–5*) redundantly inhibited the expression of the master regulator *SmcR* and modulated the expression of virulence factors in cells lacking iron at low cell density and in *Vibrio vulnificus* at high cell density [13,14]. Fis is a regulator that negatively regulates the expression of the quorum-sensing master regulator *OpaR* and inhibits capsule polysaccharide (CPS) and biofilm formation, implying that Fis plays a direct role in the quorum sensing, biofilm formation, and metabolism of *V.*
*parahaemolyticus*. Interestingly, when Fis binds directly to the regulatory regions of Qrr genes, it acts as a positive regulator [15].

*Vibrio parahaemolyticus* is a gram-negative bacterium generally isolated from estuarine and marine sources, and its infection leads to gastroenteritis related to seafood consumption in humans worldwide [16,17]. Nevertheless, the pathogenesis of *V. parahaemolyticus* infection has not been completely and explicitly elucidated. The major virulence factors that have been recently characterized include adhesins, flagella, hemolysin, and two type III secretion systems (T3SSs), namely T3SS1 and T3SS2 [18,19]. Every T3SS produces a distinct group of effectors that contributes to virulence by interacting with various host targets and performing various functions. T3SS1 is responsible for cytotoxicity, whereas T3SS2 is primarily responsible for enterotoxicity [20,21,22,23]. Five Qrrs have been identified in *V. parahaemolyticus*, but only the functions of *Qrr2*–*Qrr4* have been elucidated. They activate the target gene *AphA* at a low density and promote the synthesis of the low-density regulator *AphA*. They can also act on the target gene *OpaR* and inhibit the production of the high-density regulator *OpaR* [24,25]. However, the regulation of toxicity by Qrrs in *V. parahaemolyticus* has not been directly proven. A previous study reported that *AphA* can positively regulate the expression of T3SS1 to affect virulence, whereas *OpaR* negatively regulates the expression of T3SS1 [24,26]. T3SS is an injectable needle-like structure consisting of multiple distinct structural proteins that secrete virulence effectors into host cells to cause cellular damage, and T3SS1 mainly causes a cytotoxic effect on the host [17,25,27]. Therefore, it was speculated that the Qrrs of *V. parahaemolyticus* might regulate the expression of T3SS1 through the quorum sensing pathway to regulate the pathogenicity of the bacteria in the host. Subsequently, in a transcriptome analysis of *V. parahaemolyticus*-infected human HeLa cells, it was found that the main effectors of T3SS1 were significantly up-regulated at all time points post-infection (2, 3, 4, 6, and 8 h). However, the expression level of *AphA* did not change significantly upon infection. Similarly, the expression of *OpaR* did not change significantly during the early period of infection, and a down-regulated expression trend was observed only during the later period of infection [28,29]. Most studies have focused on the regulation of Qrrs associated with quorum-sensing genes [30,31,32]. These studies indicate that Qrrs of *V. parahaemolyticus* play a vital role in the regulation of virulence genes. However, the role of Qrrs in the virulence of *V. parahaemolyticus* remains unclear.

In this study, we detected the relative mRNA expression levels of Qrrs during the infection of Caco-2 cells with *V. parahaemolyticus*. One of the Qrrs, *Qrr5*, showed significant changes in expression during the infection. The *Qrr5* mutant strain was utilized to infect Caco-2 cells, and transcriptome analyses were performed during infection. We aimed to unveil the mechanism of Qrr sRNAs in regulating virulence factors in *V. parahaemolyticus.* The findings of this study are expected to promote the development of effective methods to alleviate the toxicity of this pathogen.

## 2. Materials and Methods

### 2.1. Bacterial Strains, Plasmid, and Growth Conditions

*Vibrio parahaemolyticus* RIMD2210633 and derivatives were cultured in 3% NaCl (*w*/*v*) alkaline peptone water or on chromogenic *Vibrio* agar (Huankai, Guangzhou, China) at 37 °C. *Escherichia coli* strain SM17-λ-pir (Stored in our laboratory, Guangzhou, China) was used for plasmid transformation and conjugation.

The concentrations of antibiotics: ampicillin (100 μg/mL); chloramphenicol (34 μg/mL) for *E. coli*; chloramphenicol (5 μg/mL) for *V. parahaemolyticus*.

### 2.2. Mutant and Complementary Strain Construction

The mutant *V. parahaemolyticus* strain was constructed by deleting the *Qrr5* gene, as previously described using *SacB*-based allelic exchanges with plasmid pDS132 [33]. In brief, for the construction of *Δ**Qrr5*, primer sets of *Qrr5*-U-F/*Qrr5*-U-R and *Qrr5*-D-F/*Qrr5*-D-R were used to amplify the upstream and downstream sections of *Qrr5*. A 20-bp overlap was added to each PCR fragment amplified by *Qrr5*-D-R/*Qrr5*-U-F, allowing the second fragment of approximately 1200-bp, including a 600-bp fragment upstream and about a 600-bp fragment downstream of *Qrr5*, using the primers *Qrr5*-U-F and *Qrr5*-D-R, respectively. The fused fragment was ligated into SphI- and SacI-digested plasmid pDS132 using the In-Fusion HD Cloning kit (Takara, Beijing, China) according to the protocol. The constructed plasmid was transformed into *V. parahaemolyticus* cells by *E. coli* S17-λ-pir. Double cross-over mutants were selected on 10% sucrose LB agar plates. The deletion mutants were verified by PCR and sequencing. The complementation of mutants was constructed using plasmid pBAD33, and the *Qrr5* fragment with its promoter was ligated into SacI and XbaI-digested pBAD33. Next, pBAD33-*Qrr5* was transformed into the mutant strain using *E. coli* S17-λ-pir. Primers used for plasmid and mutant construction are listed in Table 1 and Appendix A.

### 2.3. RNA Secondary Structure Prediction

The secondary structure of RNA was predicted with machine learning using mFold software with machine learning from the UNAFold Web at http://www.unafold.org/ (accessed on 12 October 2021) [34].

### 2.4. Measurement of Growth Curves

To measure growth curves, all strains were inoculated into a 96-well microplate (OD_600_ = 0.05). Growth curves were obtained by determining the absorbance of each well at 600 nm every 30 min for 24 h at 37 °C utilizing a microplate spectrophotometer (EPOCH2, Bio Tek, Winooski, Vermont, USA), and data were recorded utilizing Gen 5 (EPOCH2).

### 2.5. Measurement of Cytotoxicity and Adherence

Caco-2 cells were infected with *V. parahaemolyticus* as previously described [35]. Briefly, Caco-2 cells at approximately 1 × 10^5^ cells/well were incubated in Dulbecco’s modified Eagle medium (DMEM) (Gibco, ThermoFisher Scientific, Shanghai, China) containing 10% (*v*/*v*) fetal cow-like serum and 4.5 g/L D-glucose in 12-well culture dishes (Nunc, Roskilde, Denmark) at 37 °C, under anaerobic conditions with 5% CO_2_. The cytotoxicity of the wild-type, mutant, and complement strains was detected by estimating the activity of cytoplasmic lactate dehydrogenase (LDH) leaking from Caco-2 cells. Caco-2 cells were infected with wild-type, mutant, and complement strains (1 × 10^7^ CFU/mL) under anaerobic conditions for 2, 4, and 6 h (multiplicity of infection, MOI = 100), and an LDH assay was performed to quantify cytotoxicity, in accordance with the manufacturer’s instructions (Solarbio, Beijing, China).

Trypsin-digested Caco-2 cells (10^5^ cells/well) were mixed thoroughly with DMEM- resuspended bacterial solution (10^7^ CFU/mL) at a volume ratio of 1:1 and then co-cultured under anaerobic conditions with 5% CO_2_ for 1 h. Next, the cells were collected via centrifugation at 1000× *g* for 10 min and washed twice with phosphate-buffered saline (PBS) (10 mM, pH 7.4) to remove nonadherent bacteria. Then, 500 μL of 0.01% Triton X-100 (Solarbio, Beijing, China) was added to lyse the cells, and finally the cell lysate was gradient-diluted with PBS and counted on plates to obtain the concentration of adherent bacteria. Adherence (%) = (CFU_adherence_/CFU_before_) × 100%.

### 2.6. Measurement of Biofilm Formation

Biofilms were established utilizing the method developed by O’Toole and Kolter [36] with minor modifications. In brief, each well of a 96-well polystyrene microliter plate was inoculated with 200 μL of each culture diluted to OD_600_ = 0.2 in tryptic soy broth supplemented with 3% NaCl. The cells were then incubated at 37 °C without shaking for 48 h. After planktonic cells had been eliminated, biofilms connected to the wall were washed gently with PBS (10 mM, pH 7.4), stained with 230 μL of 0.1% (*w*/*v*) crystal violet, and kept in the dark for 20 min. Next, the floating color was eliminated, and the amount of biofilm was evaluated by eluting the crystal violet with 230 μL acetic acid (33%) and estimating the absorbance at 590 nm (OD_590_).

### 2.7. RNA-Seq of V. parahaemolyticus after Infection of Caco-2 Cells

Caco-2 cells were infected with *V. parahaemolyticus* under anaerobic conditions for 2, 4, and 6 h (MOI = 100). The culture broth was then centrifuged at 1000× *g* for 10 min to remove Caco-2 cells, resulting in a supernatant containing *V. parahaemolyticus.* Next, the Caco-2 cells were washed 4–5 times with PBS by vigorous shaking to wash off the adherent bacteria, which were then collected, and mixed with the bacteria-containing supernatant in a new tube. The bacterial mixture was then centrifuged at 5800× *g* for 15 min to obtain the total bacterial pellets. The pellets were washed three times with PBS gently (10 mM, pH 7.4) and all experiments were conducted with three biological replicates. Total RNA was extracted using RNAiso Plus (Taraka, Beijing, China), with DNase I treatment for 30 min, followed by purification with ethanol. A RiboCop rRNA depletion kit (Lexogen, Vienna, Austria) was used to remove rRNA from the samples. Successful removal of rRNA was confirmed using an Agilent 2100 bioanalyzer.

Fragmented mRNA was used as a template to synthesize the first strand of cDNA, and RNase H was used to degrade the RNA strand for preparation of the second strand of cDNA. The purified cDNA was repaired and connected to the sequencing adapter, and the library was completed after purification. AMPure XP reads were screened at approximately 370–420 bp cDNA.

### 2.8. Reading Mapping and Analysis

Read quality was first examined using FastQC [37]. The original read data from RNA-Seq libraries were then mapped to the *V. parahaemolyticus* RIMD2210633 genome, using Bowtie2 with standard parameterization. The htseq-count tool was used to quantify the reads aligned to the identified genes using gene annotations by Rockhopper. The calculated uniquely mapped read counts were fed into DESeq2 (version 1.14.1) for the quantitation of significant gene expression with standard parameterization. The differentially expressed genes (DEGs) identified by DESeq2 were filtered using a moderate absolute |log2(Foldchange)| >0 and *p*-value < 0.05. The read counts were transformed into FPKM to evaluate the gene expression levels. The Kyoto Encyclopedia of Genes and Genomes (KEGG) database was used to annotate the lists of significantly expressed genes with pathway details. The Gene ontology (GO) database predicted the GO functions of the genes in *V. parahaemolyticus*. Heat maps and boxplots were constructed using the R basic package [38].

### 2.9. Temporal Analysis

Temporal analysis was performed using Short Time-series Expression Miner (STEM) software according to the protocol at http://www.cs.cmu.edu/~jernst/stem/ (accessed on 20 December 2021) [39].

### 2.10. Weighted Gene Co-Expression Network Analysis (WGCNA)

A weighted relationship network was developed (with WGCNA) using R package ver.4.1 [40]. After mapping the transcriptome data to the reference transcripts using Kallisto software ver.0.45.0 [41], the transcripts per million (TPM) values of each gene in each sample were calculated. Gene expression analysis was performed for the wild-type and *ΔQrr5* groups, and the trend module of the distribution of the associated gene clusters in different groups was established. Given that most related genes should be in the same module, the intersection of the two groups was considered to indicate the related genes. The module to which *Qrr5* belongs should be confirmed, and an intersection analysis between the genes of this module and previous genes was conducted. A co-expression network of potential regulatory genes was established considering a co-expression weight greater than 0.5.

### 2.11. Quantitative Real-Time PCR (qRT-PCR)

qRT-PCR was performed using a two-step qPCR system (LightCycler 96, Roche, Basel, Switzerland) to determine the gene expression level. The extracted RNA was used for reverse transcription to obtain cDNA according to the protocol (Prime Script RT reagent Kit, Takara). TB Green Premix Ex Taq II (Takara) was utilized to carry out the qRT-PCR reaction. The 2^−ΔΔCt^ method was used to assess the fold change in the target genes compared to the housekeeping gene (*gyrB*). Each qRT-PCR was performed at least three times. Dataset S1 includes the primers used for qRT-PCR.

### 2.12. Statistical Analysis

A one-way ANOVA was used for comparisons among multiple groups. *p*-value < 0.01 (**); *p*-value < 0.05 (*) was considered statistically significant and *p*-value > 0.05 was non-significant (ns). GraphPad Prism ver.8.0 was used for statistical analysis.

## 3. Results and Discussion

### 3.1. Expression of Different Qrrs during the Infection of V. parahaemolyticus RIMD2210633 to Caco-2 Cells 

It has been found that *Qrr1-5* negatively regulate the high cell density quorum-sensing regulator *OpaR*, which is required for biofilm formation and acts as a repressor of swarming motility [15,33]. However, we found that the cytotoxicity of Caco-2 cells increased with increasing post-infection time, and *Qrr5* was significantly upregulated at different post-infection time points compared to *Qrr1*–*4* (Figure 1a,b). A previous study found that only *Qrr5* was significantly upregulated when *V. parahaemolyticus* infested the host, while the expression of *Qrr1*–*Qrr4* showed no change at all time points post-infection, which is completely consistent with our findings [42].

In the functional analysis of *Qrr5*, we found that *Qrr5* was located in chromosome 1 from 1,728,021 to 1,728,129 in *V. parahaemolyticus* RIMD2210633 (Appendix A). The secondary structure of *Qrr5* was portrayed according to its sequence, and the dG was −31.1 (Appendix A).

### 3.2. Qrr5 Is Required for Biofilm Formation, Cytotoxicity, and Adhesion to Caco-2 Cells in V. parahaemolyticus RIMD2210633

To examine the function of *Qrr5* in *V. parahaemolyticus*, *Qrr5* was deleted from the *V. parahaemolyticus* RIMD2210633 genome, and the mutants *ΔQrr5* and *ΔQrr5*::*Qrr5* were obtained and verified (Appendix A). In order to compare the difference in bacterial growth, we measured the OD_600_ after culture for 16 h and found no significant difference between the mutant and wild-type strain (Figure 2a). The biofilm formation of *ΔQrr5* was significantly decreased at 48 h compared to that of the wild-type strain (Figure 2b). The adhesion rate of mutant *ΔQrr5* to Caco-2 cells was also significantly lower than that of the wild-type (Figure 2c). Moreover, the cytotoxicity of Caco-2 cells infected by mutant *ΔQrr5* was significantly attenuated compared to that of the wild-type and *ΔQrr5::Qrr5* strains at different time points post-infection (Figure 2d). These data demonstrate that *Qrr5* is required for biofilm formation, cytotoxicity, and adhesion to Caco-2 cells in *V. parahaemolyticus* RIMD2210633.

### 3.3. Untargeted Analysis of Transcriptome Data of ΔQrr5-Mutant V. parahaemolyticus RIMD2210633

To identify alterations in gene expression of *V. parahaemolyticus* during infection, which could be triggered by the precise regulation of virulence by *Qrr5*, we set various time points after infection to Caco-2 cells. Using the Deseq2 program, differentially expressed genes between wildtype and *Qrr5* mutant strains (log_2_FC > 1; log_2_FC < −1 and *p*-value < 0.05) were identified during various periods (Appendix A). A core set of 540 common transcripts was found at various time points, and 185, 586, 355, and 74 transcripts were mainly altered at 0, 2, 4, and 6 h, respectively, in mutant *ΔQrr5* compared to those in the wildtype strain (Figure 3a). The numbers of downregulated and upregulated genes among all DEGs between the *ΔQrr5* and wild-type strains at various time points post-infection. (Figure 3b).

### 3.4. Targeted Transcriptome Analysis of ΔQrr5-Mutant V. parahaemolyticus RIMD2210633

Bacterial pathogens usually use various strategies to attack mammalian cells, preventing the immune system from responding and causing damage to infected tissue sites [43,44]. Pathogens frequently employ a secretion system to release virulence factors intracellularly into the host cells or extracellularly to interfere with host cell functions and promote bacterial infection [45,46]. *Vibrio parahaemolyticus* harbors several secretion systems, including T2SS, two T3SSs (T3SS1 and T3SS2), and two T6SSs. All T3SSs are associated with the toxins in mammalian hosts [16]. Reports on the function of T6SSs are limited, but it has been speculated that T6SS2 contributes to bacterial virulence [47,48]. sRNAs are post-transcriptional regulatory molecules that play an indispensable role in adjusting vital processes in bacterial physiology toward host invasion [49,50]. Analysis of the transcripts of *V. parahaemolyticus* revealed that the deletion of *Qrr5* significantly affected the expression of the T3SS1 gene cluster during the infection period, especially at 6 h post-infection (Figure 4). The gene clusters of other secretion systems, such as T3SS2, T6SS, and T2SS, were slightly upregulated during the infection period (Figure 4).

### 3.5. WGCNA of Transcriptome Data of ΔQrr5-Mutant V. parahaemolyticus RIMD2210633

To obtain an outline of the dynamic variations of DEGs during the process of infection with *V. parahaemolyticus*, we first performed a trend analysis of all genes in the wild-type and *Qrr5* knockout strains separately using STEM to determine the expression trend of T3SS1 genes in wild-type. The results revealed that most T3SS1 genes in the wild-type group were distributed in profiles 14 and 15, whereas most T3SS1 genes in the *ΔQrr5* group were distributed in profile 10 (Figure 5a,b). Based on the principle that the genes related to T3SS1 regulation should also be classified into the same profile, the genes in profiles 14 and 15 of the wild-type group were intersected with the genes in profile 10 of the *Δ**Qrr5* group, and 138 genes were obtained. The KEGG pathway enrichment results of all genes in profile 14 (the profile where *Qrr5* is located) of the wild-type group were mostly concentrated on ABC transporters and two-component systems (Figure 5c). All genes in profile 10 (the profile where most T3SS1 genes are located) of the *ΔQrr5* group mainly focused on the biosynthesis of secondary metabolism (Figure 5d). Subsequently, the expression level of *Qrr5* was used as a trait for WGCNA. All genes whose expression met the threshold were divided into 11 gene expression modules (Figure 6a). *Qrr5* was located in the turquoise module, which was also the gene module most significantly correlated with *Qrr5* expression (Figure 6b). The genes in the turquoise module were intersected with the above 138 genes, and a total of 61 intersecting genes with a potential co-expression relationship with *Qrr5* were identified, consistent with the expression trend of the T3SS1 genes. KEGG pathway enrichment analysis of the turquoise module, where *Qrr5* is located, showed that ABC transporters and two-component systems were the most related pathways (Figure 6c). Considering a co-expression weight greater than 0.5 as a potential direct regulatory relationship, a direct regulatory gene network of *Qrr5* was constructed (Figure 6d). The functions of the co-expression network genes are listed in Appendix A. The results showed that *VP2914*, *VP0485*, *VP2492*, and *VP1665* exhibited strong interactions with *Qrr5*, which encodes D-2-hydroxyacid dehydrogenase, TIGR01212 family radical SAM protein, ammonium transporter, and T3SS1 chaperone *SycN*, respectively. A novel sRNA, *EsrF*, was found to respond to ammonium transporters to sense high ammonium concentrations in the colon and promote *E. coli* O157:H7 pathogenicity by enhancing bacterial motility and adhesion to host cells. The coexistence of ammonium transporter and channel mechanisms contributed to the virulence of pathogenic fungi [4,51]. Moreover, the T3SS1 chaperone *SycN*/*YscB* interacts with the secreted protein *YopN* via an N-terminal chaperone-binding domain to subvert the defenses of mammalian hosts in *Yersinia pestis*, and it was found that they interact with both *EsaB* and *EsaM* within the bacterial cells, contributing to the pathogenesis of *Edwardsiella tarda* [52,53,54]. Our results demonstrate that *Qrr5* is an important sRNA in the regulation and the pathogenesis of *V. parahaemolyticus*. Analysis of the expression levels of Qrrs in this transcriptome revealed that *Qrr5* was significantly upregulated at all time points post-infection, whereas *Qrr1–4* expression showed no significant changes throughout, which was completely consistent with a previous finding [42]. In summary, *Qrr5* regulates cytotoxicity mainly by regulating the T3SS1 gene, as well as participating in the ABC transporters and two-component system pathways during infection.

The expression of genes involved in the gene regulatory network of *Qrr5* was further validated by qRT-PCR. The results showed that the expression patterns of these genes at 0, 2, 4, and 6 h post-infection were generally in accordance with those by RNA-Seq. At each post-infection time point, *VP1766*, *VPA1068*, *VP0485*, *VP2914*, and *VP1665* were up-regulated. Moreover, *VP2492*, *VP0486*, and *VPA0628* were up-regulated at 2 h and down-regulated at 0, 4, and 6 h post-infection (Appendix A). 

## 4. Conclusions

This study reveals that *Qrr5* is related to cytotoxicity and can significantly positively regulate virulence gene expression levels in *V. parahaemolyticus.* This study also elucidates the systematic transcriptomic profile of *V. parahaemolyticus* mutants during infection of Caco-2 cells. A total of 185, 586, 355, and 74 DEGs were obtained at 0, 2, 4, and 6 h post-infection, respectively, which were mainly related to ABC transporters and two-component system pathways. Using WGCNA, we identified 17 key genes involved in the gene regulatory network associated with *Qrr5*. These findings highlight the importance of core genes that may be regulated by *Qrr5* during infection in Caco-2 cells and provide a basis for further research on the mechanisms of *V. parahaemolyticus* infection.

## Figures and Tables

**Figure 1 microorganisms-10-02084-f001:**
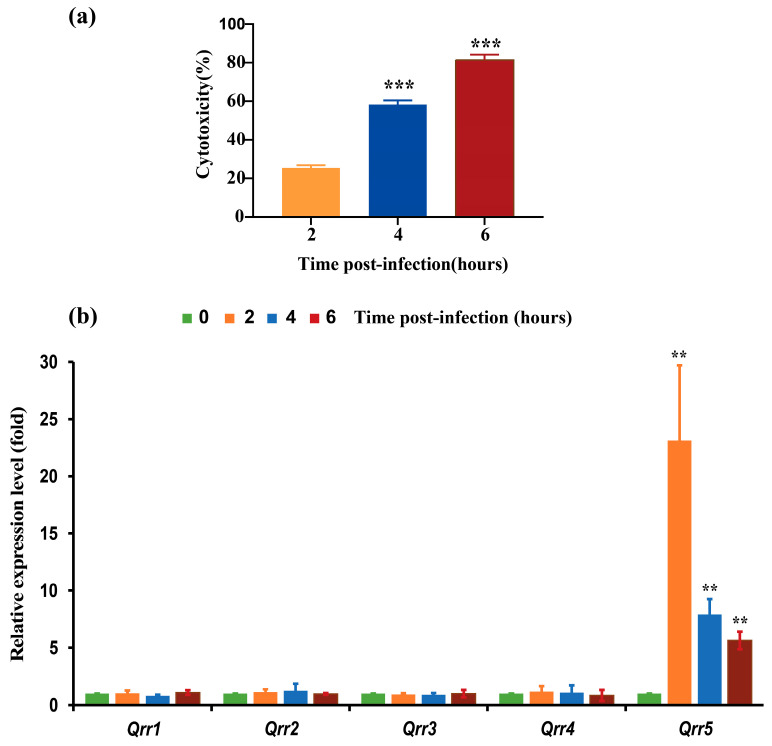
(**a**) The cytotoxicity of Caco-2 infected by *Vibrio parahaemolyticus* RIMD2210633 during various post-infection time points. (**b**) The relative mRNA expression of five sRNA Qrrs in *V. parahaemolyticus* RIMD2210633 during various post-infection time points toward Caco-2 cells. *p*-value < 0.01 (**); *p*-value < 0.001 (***).

**Figure 2 microorganisms-10-02084-f002:**
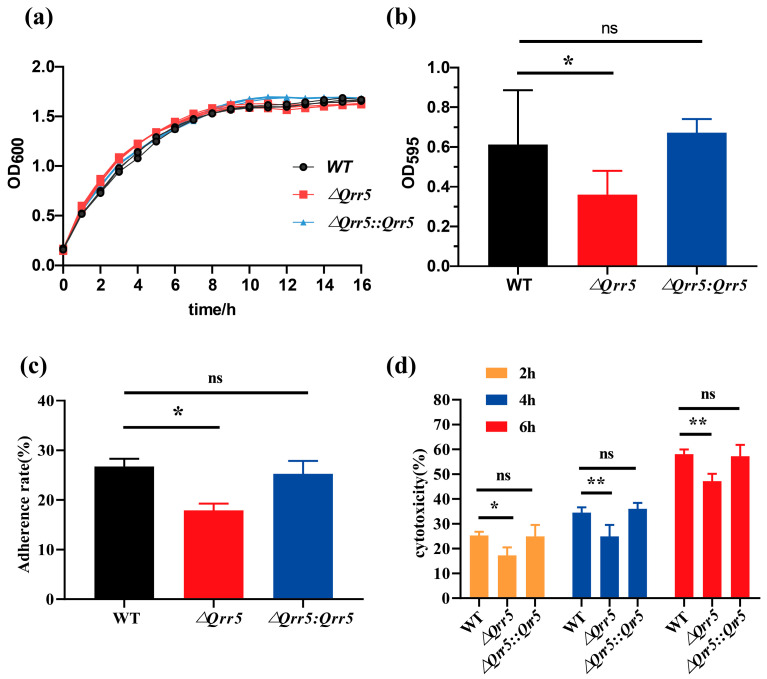
The phenotypes of *ΔQrr5* deletion mutant of *Vibrio parahaemolyticus* RIMD2210633. (**a**) The growth curves of wild-type, *ΔQrr5* deletion mutant, and the complementary strains were measured at OD_600_ for 16 h. (**b**) Biofilm formation of different strains was measured at 595 nm at 48 h by crystal violet staining. (**c**) The adherence rate of different strains to Caco-2 cells. (**d**) The cytotoxicity of Caco-2 cells (MOI = 100:1) infected by different strains at 2, 4, and 6 h post-infection. *p*-value < 0.05 (*); *p*-value < 0.01 (**); ns: not significant.

**Figure 3 microorganisms-10-02084-f003:**
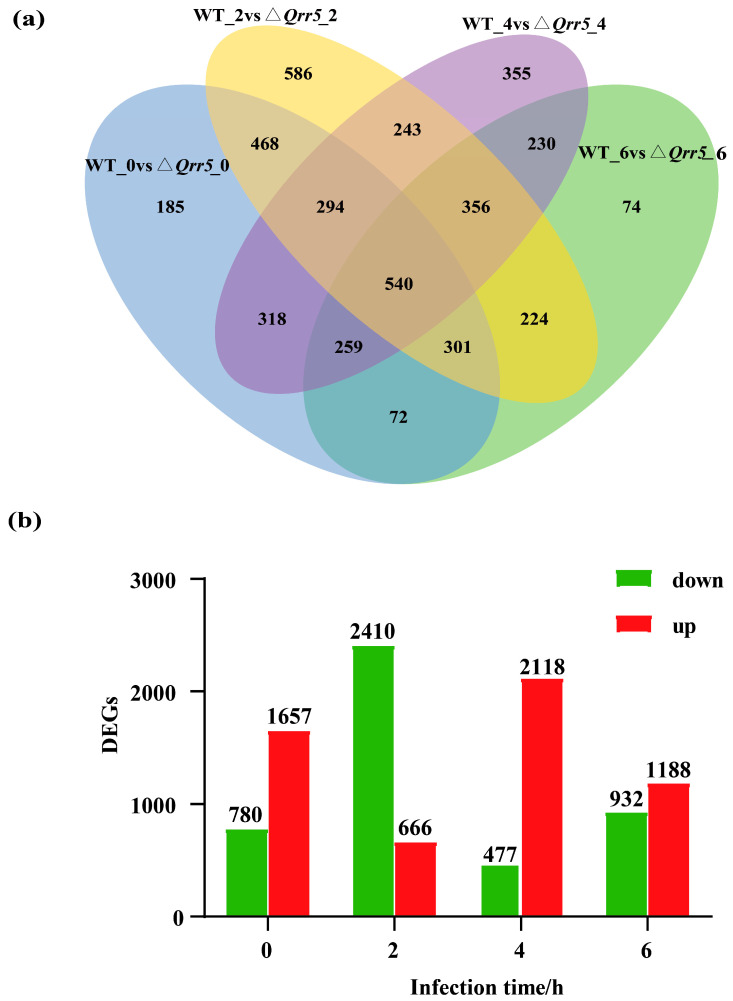
The transcription outcomes of *V. parahaemolyticus* after its infection to Caco-2 cells. (**a**) Venn diagram portraying the number of significantly expressed genes (log_2_FC > 1; log_2_FC < −1 and *p*-value < 0.05) of wild-type and *ΔQrr5* strains across various post-infection time points. (**b**) The numbers of the down-regulated and up-regulated of all differentially expressed genes in the wild-type and *ΔQrr5* strains at various infection time points.

**Figure 4 microorganisms-10-02084-f004:**
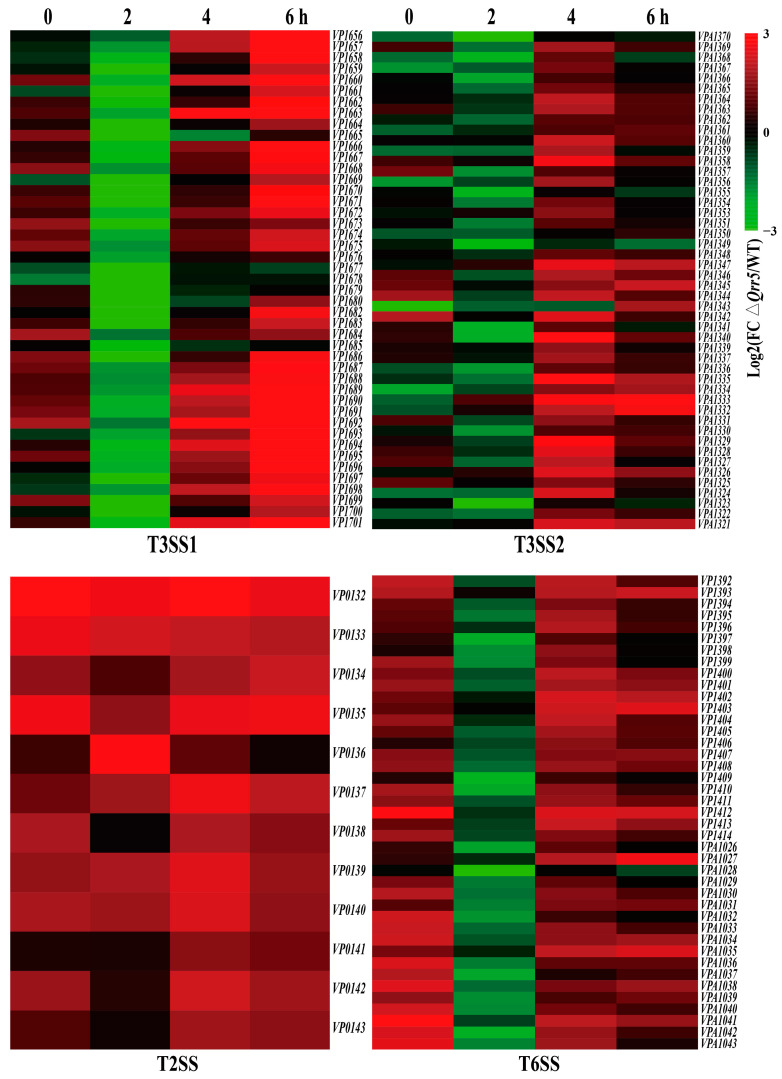
Differential expression of secretion system gene clusters in *V. parahaemolyticus* mutant *ΔQrr5* during the infection period compared to the wild-type strain (including T3SS1, T3SS2, T2SS, and T6SS gene clusters).

**Figure 5 microorganisms-10-02084-f005:**
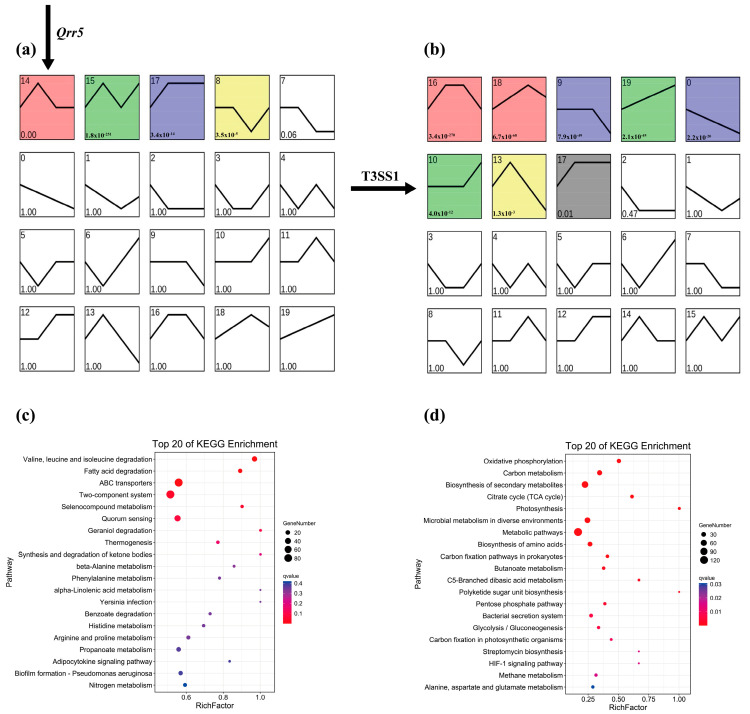
The STEM analysis of mutant *ΔQrr5* during the infection period compared to wild-type strain by RNA-Seq. (**a**) Gene trend expression enrichment map of the wild-type group. (**b**) Gene trend expression enrichment map of *ΔQrr5* group. The colors represent the module where the major genes are located, and the white represents the module where fewer genes are located. (**c**) KEGG pathway enrichment analysis results of all genes in profile 14 (the profile where *Qrr5* is located) of the wild-type group trend. (**d**) KEGG pathway enrichment analysis results of all genes in profile 10 (the profile where most T3SS1 genes are located) of the *ΔQrr5* group trend.

**Figure 6 microorganisms-10-02084-f006:**
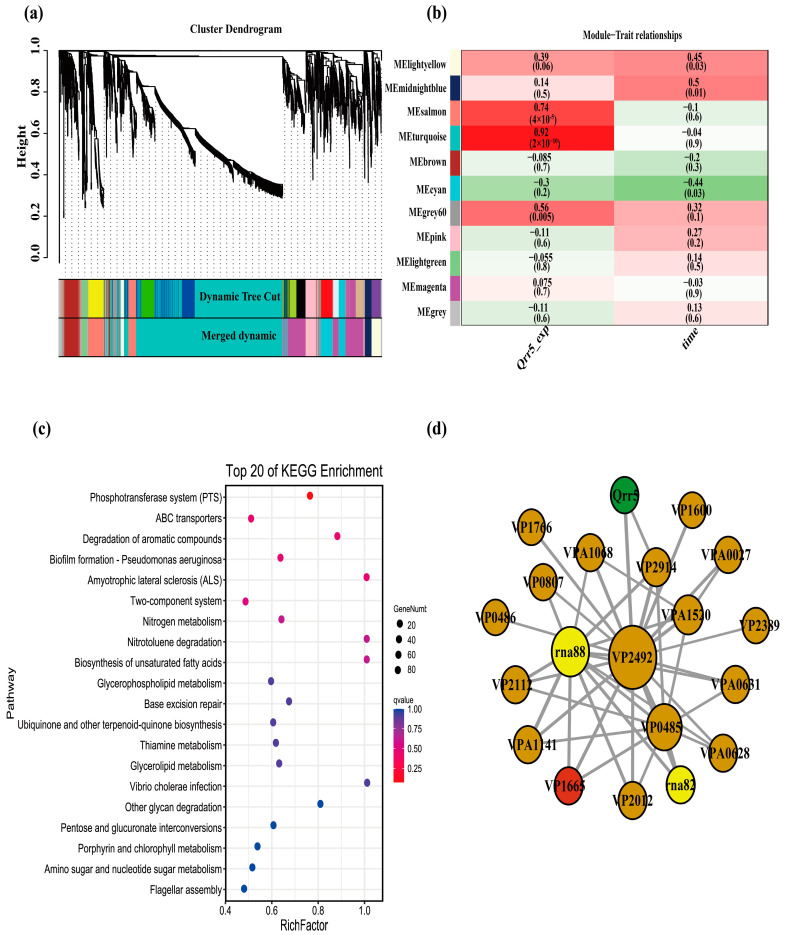
The WGCNA analysis of mutant *ΔQrr5* during the infection period compared to wild-type strain by RNA-Seq. (**a**) Hierarchical cluster tree showing eleven modules of co-expressed genes. Each of the genes is represented by a leaf in the tree, and each of the eleven modules by a major tree branch. (**b**) Module-*Qrr5* -expression/times correlations and corresponding *P*-values. (**c**) The KEGG pathway enrichment analysis results of the turquoise module where *Qrr5* is located by WGCNA. (**d**) The co-expression network of related genes is regulated by *Qrr5*. (The red dot was the T3SS1 gene).

**Table 1 microorganisms-10-02084-t001:** Bacterial strains and plasmids were used in this study.

Strain or Plasmid	Characteristics	Reference
** *V. parahaemolyticus RIMD 2210633* **	Parent strain, serotype O3:K6	[33]
*ΔQrr5* mutant	RIMD 2210633 *ΔQrr5*	This study
*ΔQrr5::Qrr5*	pBAD33-*Qrr5* into *ΔQrr5* mutant	This study
** *E. coli SM17 λpir* **	TpR SmR recA, thi, pro, hsdR-M+RP4:2-Tc:Mu:Km Tn7λpir	[33]
pDS132	R6K ori, mobRP4 *SacB* Cm^r^	[33]
pBAD33	pBR322 ori, Amp^r^	This study

## Data Availability

Not applicable.

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
