# Peer review of "Gene Regulatory Network of the Noncoding RNA Qrr5 Involved in the Cytotoxicity of Vibrio parahaemolyticus during Infection"

_microorganisms, 2022, doi:10.3390/microorganisms10102084_

Round 1

Reviewer 1 Report (New Reviewer)

The manuscript describes functions of Vibrio parahaemolyticus Qrr5 non-coding RNA in cytotoxicity and regulatory functions.

The study is well designed and performed. The text is consistent and understandable. However, there are drawbacks on English grammar and stylistics. I recommend the authors to use professional English editing and proofreading service to check and correct the text.

Major comments:

1)      The two very important parts, Abstract and Conclusion should be rewritten to be clear. Underline the most important points and significance of the study. It should be in correct and intelligible English.

2)      There should be consistent and correct designation of genes (DNA), RNA and proteins.

3)      All symbols which appear in Figures should be explained in legends.

4)      Table 1 is probably not in correct format

5)      Problematic parts, mistakes and parts with comments are highlighted in the attached file. Please, handle all comments meticulously.

Author Response

Reviewer 2 Report (New Reviewer)

It needs to be revised and resubmitted.

Author Response

Reviewer 3 Report (New Reviewer)

In this article, the authors have elucidated the gene regulatory network involving the non-coding RNA Qrr5 in Vibrio parahaemolyticus. They concluded that Qrr5 plays an important role in cytotoxicity and adherence during infection. In addition, the authors identified a number of key genes, which are involved in the gene regulatory network associated with Qrr5. The authors have done justice to the background of the study. The manuscript is written well and nicely organized though I have some comments to improve the quality of the manuscript.

Comments

L35: EsrF à what is EsrF? Please indicate it

L46: Δfis mutant: what is fis gene?

L46: OpaR: What is the function of it?

L47: CPS? Full form

L46-49: Need to re-write the sentence

The first paragraph of introduction needs to be more organized

Mention Table 1 in the text under ‘Materials and Methods’

Figure 1: The Qrr1 – 4 expression is quite low compared to that of qrr5. Is it comparable with qrr5 expression levels in other studies in Vibrio? It is better to discuss it in the text.

Figure S2, S3, S4 à Not mentioned in the text.

Figure S2 legend: confusion DNA fragment?

L321: E. tarda à please expand it at the first time

L353: Please re-write it

Round 2

Reviewer 1 Report (New Reviewer)

The manuscript was improved substantially. I thank the authors for the efforts.

This manuscript is a resubmission of an earlier submission. The following is a list of the peer review reports and author responses from that submission.

Round 1

Reviewer 1 Report

The study looks at the potential role of sRNAs (qrr1-5) in the infection process. First, they show that qrr5 specifically is induced when Vibrio parahaemolyticus is in contact with Caco-2 cells. They then demonstrate that a qrr5 mutant has a phenotype in various processes of relevance to infection/pathogenicity. This is followed up by a transcriptome analysis comparing wt and qrr5 mutant and bioinformatics analysis. This leads the author to claim that qrr5 is required (or important) for infection. Generally, I think that the language is good, but the clarity of the analyses and the implications of the results are lacking. As I read it, and perhaps I misunderstand parts, I really do not understand the basis for so strongly stating the importance of qrr5 and determining a set of key genes required for virulence. I do not think that the data backs these claims. Overall I would advocate rejection or major revision to clarify the points below.

Major comments:

You see a great phenotype upon knocking out the qrr5 apparently affecting I suppose at least half of all genes. Now you claim this has a specific function in infection, but comparing the number of DEGS you have similar numbers at time 0 and 2 h and then less are differentially expressed at 4 and 6 h. In fig. 1 it is given that only qrr5 seems to respond to interaction with caco-2 cells. However, there seem to be a reverse correlation between expression level (high at 2 h then falling) and cytotoxicity (low at 2 h then increasing). You seem to not consider this?

You then go on to claim that qrr5 is required for biofilm. I think you should downtone that statement, as both crystal violet staining, adherences and cytotoxicity seems to go down a bit, but certainly is not abolished in the mutant. So at best it would seem to modulate the process, maybe fine-tuning? You should reflect on that I think.

In l. 256-7 again. T3SS1 genes affected esp in 6h condition where qrr5 levels are lower. In this analysis (fig 4), actually, why do you target the secretion systems specifically? I suppose with so many genes affected many other systems would be likely to be affected in one way or the other? Since interested in infection/virulence, you could expand to look at the change in expression profile of all known virulence factors?

Fig 3 is unclear to me. What are you actually showing, is it just whether you see a given transcript in the condition or that it is significantly different between the wt and qrr5 mutant? You wording is unclear l .233 “genes with different expressions” and l. 240 “significantly expressed genes”. Please rephrase to make it clear what I am looking at.

So since I do not know, then my commenting is thereafter. If we are looking at transcripts just identified in all conditions it might not be strange with an overlap of 1070 bp. If on the other hand it is indeed differentially expressed genes then to have 1070 genes different in all conditions seems like a lot? In fact, one could say that it most be so greatly impacted the cell, that probably most cellular processes are perturbed in one way or the other. Consequently, it might be necessary with more direct evidence that qrr5 plays a role in infection. I.e. to demonstrate functional, regulatory interactions in the lab.

In the same instance, you make a PCA analysis, but why and what do you conclude from it. Guide the reader, what are we seeing here and what is the significance of this analysis? Simply stating that a PCA analysis was conducted (l. 234-237) is not good enough I think.

For the Vibrio cell to impose an effect on the Caco2 cells, I surmise that they would need to be in close contact. In figure 2, you actually talk about adherence. This experiment should be explained in materials and methods section!

Further, you conclude that the qrr5 exerts its effect via the T3SS which I suppose even more so would necessitate close contact.

Now for your experimental approach, as I understand it, you have a layer of caco2 cells upon which you inoculate the bacteria (in liquid on top of the caco2 cells). Is that correct? If so, then I suppose that a proportion of bacteria could interact with the caco2 cells and another proportion could swim in the medium above. When you separate the cells by centrifugation, would you not actually loose the Vibrio cells that adhere to Caco cells (and hence are the most relevant to study)? So if there is a change in expression profile due to the interaction, then you are selecting for those that are not in contact  and hence not the population you wish to study? I think it would be quite relevant to have a control reaction where you have the WT in contact with Caco2 or not (but in same liquid medium) if you wish to claim that the targets are relevant for infection.

Of course, comparing wt and qrr5 mutant is possible, but maybe it is not relevant for infection.

Generally speaking it is very difficult to follow section 3.5 where you have the WGCNA analysis and some other approach to group the genes. You need to put more emphasis on explaining what the analysis is intending to show, and guide the reader to understand what it then shows and the significance thereof. As it is now, it seems the analyses are made mainly for the sake of the analyses. In fig 5c+d. In one case you have the genes in the group with qrr5 and in the other the selection criterium is that it contains majority of T3SS1 genes. Would it not then be more reasonable to compare the T3SS1 enriched profile(s) in both (i.e. pool profile 14-15 as I understood)?

I think that looking at the gene expression profiles seems like an interesting approach to find co-regulated genes, but what else do you find in these groups (i.e. non T3SS1 genes) that would then putatively be regulated by qrr5?

In contrast, what are the insights we obtain from the KEGG analysis, this is more unclear to me?

When considering Fig 5 and 6, it seems that both have a KEGG profile of a group of genes that contain qrr5 and it is not the same profile – so two different ways of grouping the genes? What is the relevance of using these two distinct approaches and why do you think the outcome is different?

What are we seeing in fig 6d. You call it a co-expression network and have lines between the genes (looks like a string analysis) – what are the significance of the lines and how where they generated? Does it indicate actual interactions between the proteins that could infer functionality?.Also, please explain the colorcoding.

Overall, I think you need to re-think section 3.5 to make it much clearer and perhaps also reflect on whether you need to have what appears to be redundant analyses (and if so, make it clear why).

Minor comments:

l. 22. DEGs exhibited a dynamic change in expression – what does that mean? Please clarify or revise.

l. 21. How many genes are in V. parahaemolyticus?? Are you really claiming that 3500 genes were differentially expressed when comparing WT and qrr5 mutant?

l. 23. Accordingly. So to state that because of differential expression (if that is what you mean), you find 17 T3ASS genes in WGCNA?

l. 27: You really need to clarify how the presented data can be seen as a source of target virulence factors.

l. 28. You look from perspective of the bacterium, so how can you say much about the mammalian cells immune response during infection??

Fig1. Is it not surprising that you see no effect whatsoever for qrr1-4 especially since, as I understood from the intro, the qrr2-4 are the one known to play a role in regulation various transcriptional regulators? Is this contrasting other studies, if so perhaps discuss this further. Is there a risk that the result could be an artefact?

l. 266-272. You say that qrr5 is in goup 14 so peaks at 2 h. and that other T3SS1 genes are in that group, but this does not seem consistent with figure 4 where expression of T3SS1 is at it highest in 6h (perhaps with exception of two genes)?

L.323 5021 DEGs: are you counting the same genes multiple times? You really need to state the number of genes in the strains studied.

L. 327. So a core set of 17 key genes , are they the ones in fig 6d (brown +red)? And really, I think it is an over-interpretation to call this a core set of genes in virulence. You have as I see it, multiple groups of differential expression profiles and you then picked the ones containing T3SS1 and claimed them to be the relevant genes. But there is no actual experimental proof of causality. This is a big weakness of this study.

The manuscript is generally very well written, but I found a few places that could be revised:

l. 17 ..is of great significance foodborne pathogen..

l. 22 ….when the loss of Qrr5.

And likely more. Language is definitely good, but perhaps go through one more time.

Reviewer 2 Report

The purpose of this article was to evaluate the effect of noncoding RNA Qrr5 involved in the cytotoxicity of Vibrio parahaemolyticus during infection. It is an interesting study, but I have some questions for the authors?

Why was a particular tumor line chosen?

In addition to the CaCo-2 cell line, why didn't you also use a normal cell line?

Why were the cells maintained in anaerobiosis, what was the reason? Did anaerobiosis not influence cytotoxicity? From a morphological point of view and related to the adhesion capacity of the cells, were changes identified?

Round 2

Reviewer 1 Report

All in all, the authors did not convince me that the manuscript is ready for publication. As I have now looked at it twice, I do not wish to see it again. I advocate rejection.

As noted in my original review, I advocated rejection or major revision. I had multiple comments and while some of them led to actual changes in the manuscript, more effort was put into trying to explain why it was already okay. In most cases, I did not find that very convincing.

There was a recalculation of the DEG data that led to completely new numbers. This is of course lucky that this was picked up. However, this led to an even smaller core of unchanged genes – so even a bigger fraction of the cells genes are changed in response to the deletion. If understanding correctly, more than 2/3 of all detected genes (RNAs) were differentially regulated – but still the role is specifically in infection?? Makes little sense to me that this should be the case. So I still fear that you might over-interpret the role of qrr5 in infection.

Also the number of genes are not consistent

L. 256 core set of 1070 transcripts (but according to recalc should be 540?)

DEGs: 185+586+355+74 = 1200

Fig 3b: for each time point you have more genes if summing up + down.

So there is definitely some issue there.

And well, even if you do not agree, then I think section 3.5 is not well written and clear.

In section 2.9 you reference a homepage but might fail to ref the original research. More details could be given (in case one day the homepage is no longer accessible).

I could provide more comments, but I think it is futile as the authors did little with my initial comments.

Reviewer 2 Report

The article has been substantially corrected, after the final correction it can be considered for publication.